# Detection of a Novel Gull-like Clade of Newcastle Disease Virus and H3N8 Avian Influenza Virus in the Arctic Region of Russia (Taimyr Peninsula)

**DOI:** 10.3390/v17070955

**Published:** 2025-07-07

**Authors:** Anastasiya Derko, Nikita Dubovitskiy, Alexander Prokudin, Junki Mine, Ryota Tsunekuni, Yuko Uchida, Takehiko Saito, Nikita Kasianov, Arina Loginova, Ivan Sobolev, Sachin Kumar, Alexander Shestopalov, Kirill Sharshov

**Affiliations:** 1Federal Research Center of Fundamental and Translational Medicine, 630060 Novosibirsk, Russia; nikitadubovitskiy@gmail.com (N.D.); nauka@duck.com (N.K.); loginova995@gmail.com (A.L.); sobolev.riov@yandex.ru (I.S.); shestopalov2@mail.ru (A.S.); sharshov@yandex.ru (K.S.); 2Federal Research Center “Krasnoyarsk Science Center of the Siberian Branch of the Russian Academy of Sciences”, Research Institute of Agriculture and Ecology of the Arctic, 663300 Norilsk, Russia; al.prokudin@mail.ru; 3Division of Transboundary Animal Disease, National Institute of Animal Health, Tsukuba 305-0856, Japan; minejun84032@affrc.go.jp (J.M.); uchiyu@affrc.go.jp (Y.U.); taksaito@affrc.go.jp (T.S.); 4Department of Biosciences and Bioengineering, Indian Institute of Technology Guwahati, Guwahati 781039, India; sachinku@iitg.ac.in; 5Department of Natural Sciences, Novosibirsk State University, 630090 Novosibirsk, Russia

**Keywords:** *Orthoavulavirus javaense*, avian paramyxovirus, APMV-1, NDV, Taimyr Peninsula, Arctic, avian influenza virus, H3N8, herring gull, northern pintail

## Abstract

Wild waterbirds are circulating important RNA viruses, such as avian coronaviruses, avian astroviruses, avian influenza viruses, and avian paramyxoviruses. Waterbird migration routes cover vast territories both within and between continents. The breeding grounds of many species are in the Arctic, but research into this region is rare. This study reports the first Newcastle disease virus (NDV) detection in Arctic Russia. As a result of a five-year study (from 2019 to 2023) of avian paramyxoviruses and avian influenza viruses in wild waterbirds of the Taimyr Peninsula, whole-genome sequences of NDV and H3N8 were obtained. The resulting influenza virus isolate was phylogenetically related to viruses that circulated between 2021 and 2023 in Eurasia, Siberia, and Asia. All NDV sequences were obtained from the Herring gull, and other gull sequences formed a separate gull-like clade in the sub-genotype I.1.2.1, Class II. This may indirectly indicate that different NDV variants adapt to more host species than is commonly believed. Further surveillance of other gull species may help to test the hypothesis of putative gull-specific NDV lineage and better understand their role in the evolution and global spread of NDV.

## 1. Introduction

*Orthoavulavirus javaense*, also recently known as avian paramyxovirus 1 (APMV-1) or Newcastle disease virus (NDV, used hereafter for this paper), is an enveloped negative-sense RNA virus in the Paramyxoviridae family. This family has been described as one of the families of RNA viruses with the potential to switch hosts rapidly [1].

The *Orthoavulavirus javaense* species is represented by genetically heterogeneous variants that are divided into two lineages (Class I and Class II) with multiple genotypes and sub-genotypes [2]. The diversity of genotypes results in a diversity of pathogenicity variants and host species. Highly pathogenic variants cause Newcastle disease in many wild and domestic bird species. Low-pathogenic variants cause subclinical infections or mild respiratory disease and are found in birds worldwide [3].

The diversity of susceptible species facilitates the global circulation of NDV [4]. No natural reservoirs have been identified for all NDV genotypes, but wild waterbirds are thought to play an important role. During routine surveillance, low pathogenic isolates are often found in *Anseriformes* and *Charadriiformes* in the same locations and periods as AIV (avian influenza virus) [4,5,6]. There are also known cases of detection of pathogenic NDV isolates in double-crested cormorants and herring gulls [7,8].

The influenza A virus (*Alphainfluenzavirus influenzae*, family *Orthomyxoviridae*) [9] is an enveloped virus with a segmented RNA genome. The influenza A virus infects humans, horses, pigs, dogs, cats, marine mammals, bats, and birds [10,11,12,13]. Natural reservoirs of influenza A virus are wild waterbirds. The global circulation of this virus, like NDV, also includes domestic birds [14,15]. Developed trade and the peculiarities of keeping poultry, along with vaccination, create special conditions for the spread and evolution of viruses. But since all subtypes of the influenza A virus infecting mammals and poultry have directly or indirectly originated from wild waterbirds [16], monitoring viruses in this natural reservoir is still an important part of epidemiological research. A study of the spread of H5N1 showed that the spread of this subtype around the world was associated with both the trade in wild and domestic birds and the seasonal migrations of wild waterbirds [15].

The migration routes of waterbirds are extensive and include the Arctic [17]. Many waterbird species migrate seasonally to the Arctic regions for breeding. Nesting sites are associated with high population density and immuno-naïve juveniles. These factors may create special conditions for the spread and evolution of viruses. Polar regions have a number of specific characteristics related to weather and seasonality, remoteness from other areas, and high levels of animal endemism [18]. Marine mammals and migratory birds are the major changing components of polar ecosystems. Polar regions could become new frontiers for the emergence and spread of significant viral pathogens. The worldwide epizootic of highly pathogenic H5N1, which began in Asia in 1996, reached Antarctica in 2023 [19,20,21]. The consequences of the mass death of marine mammals and birds and the continued circulation of highly pathogenic influenza in the Antarctic ecosystem have yet to be assessed.

The Arctic is poorly studied, with particularly few data on NDV and AIV. There is especially little information about the Arctic region of Russia [22]. Insufficient information on infectious agents in the Arctic limits understanding of the genetic diversity and evolution of viruses with a potential impact on agri-livestock farming and especially the poultry industry and on species biodiversity conservation in general.

Here, we report the results of a five-year study of NDV and AIV in waterbirds on the Taimyr Peninsula (Arctic region of Russia).

## 2. Materials and Methods

### 2.1. Sampling

The collection of biological material from birds was carried out as part of the annual monitoring of the influenza A virus on the Taimyr Peninsula (Krasnoyarsk region, Russia) (Figure 1). Samples were collected during the state hunting season under a license from the regional ministry of ecology and natural resources within the framework of the Program for the Study of Infectious Diseases of Wild Animals of the Federal Research Center of Fundamental and Translational Medicine (FRC FTM), Novosibirsk.

Cloacal swabs from wild birds were collected during the official hunting season in 2 mL tubes filled with 1 mL viral transport medium [23]. Samples were stored and transported in liquid nitrogen following the WHO guidelines [23].

The present study was conducted as per the approval and requirements of the Biomedical Ethics Committee of the FRC FTM, Novosibirsk (protocol Nos. 2013-23 and 2021-10). The study utilized the Biosafety Level-3 (BSL-3) facilities of the FRC FTM.

### 2.2. Virus Isolation

For the isolation of AIV and APMV (avian paramyxoviruses) in 10-day specific pathogen-free (SPF) embryonated chicken eggs, samples were prepared according to a standard protocol [24]. All virus isolation work was performed in the BSL-3 laboratory of the FRC FTM. After cultivation, 2 mL of allantoic fluid from each embryo infected with one sample was extracted and used for a hemagglutination test (HA) with 5% chicken red blood cells [25]. All HA-positive samples were aliquoted for AIV M and APMV RDRP gene PCR testing.

### 2.3. Avian Paramyxovirus and Avian Influenza Virus Detection

RNA was extracted from allantoic fluid samples using a kit for nucleic acid extraction (Medical-Biological Union LLC, Novosibirsk, Russia) following the manufacturer’s protocol. The presence of conservative M gene regions of the influenza A virus was determined by real-time PCR using the Influenza A Virus Real-Time RT-PCR Kit (Medical-Biological Union LLC, Novosibirsk, Russia), adapted to detect both human and avian influenza virus.

HA-positive allantoic fluid was also tested for the presence of APMV by PCR. For this purpose, cDNA was synthesized using the REVERTA-L Kit (AmpliSens, Moscow, Russia). To detect avian paramyxoviruses, family-wide oligonucleotides (PMX1 5′-GAR-GGI-YII-TGY-CAR-AAR-NTN-TGG-AC-3′ and PMX2 5′-TIA-YIG-CWA-TIR-IYT-GRT-TRT-CNC-C-3′) specific to domain III of the RNA-dependent RNA polymerase gene were used [26]. Oligonucleotides were diluted to a concentration of 50 pmol/μL. A reaction mixture was prepared using 25 μL of HS-Taq PCR-Color (2×) Master Mix (Biolabmix, Novosibirsk, Russia), 1 μL of forward and reverse oligonucleotides, and 5 μL of cDNA. Water was then added to achieve a final volume of 50 µL. The reaction mixture was incubated at 95 °C for 5 min, then for 40 cycles at 95 °C for 15 s, at 41 °C for 30 s, at 72 °C for 15 s, and then a final extension at 72 °C for 7 min.

The reaction products were visualized by electrophoresis in 1.5% agarose gel in the gel documentation system “E-Box CX5” (VILBER, Eberhardzell, Germany). A 100 bp DNA ladder DNA marker Step100 (Biolabmix, Novosibirsk, Russia) was used to estimate the amplicon size. Samples in which amplicons of the target size were detected were prepared for whole-genome sequencing.

### 2.4. Sequencing

Avian paramyxovirus sequences were obtained at the National Institute of Animal Health, Japan, using a MiSeq genome sequencer (Illumina, San Diego, CA, USA), as described previously [27]. Briefly, RNA was isolated from allantoic fluid using the RNeasy Mini Kit (Qiagen, Hilden, Germany). cDNA libraries were prepared using the NEBNext Ultra RNA Library Prep Kit (New England Biolabs, Ipswich, MA, USA). Sequencing was performed using the MiSeq Reagent Kit v.2 (Illumina, San Diego, CA, USA). Consensus sequences were constructed using Workbench software v.12.0.2 (Qiagen, Hilden, Germany).

The whole-genome sequence of the influenza A virus was obtained using the Illumina MiSeq platform (Illumina, San Diego, CA, USA) and the corresponding reagent kits according to the manufacturer’s methodology. Sequencing was performed at FRC FTM, Russia. RNA was extracted using the QIAamp Viral RNA Mini Kit (Qiagen, Hilden, Germany). Whole-genome amplification was performed using a modified protocol [28]. DNA libraries were prepared using the Nextera DNA Flex Library Prep Kit (Illumina, San Diego, CA, USA) and sequenced using the MiSeq Reagent Kit v3 (600 cycles) (Illumina, San Diego, CA, USA). Consensus sequences were generated using Bowtie software (version 1.3.1) [29].

### 2.5. Phylogenetic Analysis

#### 2.5.1. Newcastle Disease Virus

For phylogenetic analysis, the complete CDS F gene (1662 nt) of the obtained isolates and the most closely related sequences found in BLAST, as well as from the dataset described by Dimitrov et al., were used [2]. For phylogenetic analysis of whole genomes, the most closely related sequences found in BLAST and sequences from Class I isolates as an outgroup were used. Sequence alignments were obtained using the MUSCLE alignment algorithm. The phylogenetic tree was constructed using the maximum likelihood method based on the General Time Reversible model with a discrete gamma distribution (+G) and allowing for invariant sites (+I) with statistical analysis based on 1000 bootstrap replicates, as implemented in MEGA X [30]. The tree was drawn to scale, with branch lengths measured in the number of substitutions per site.

#### 2.5.2. Avian Influenza Virus

For each of the obtained genome segment sequences, a search for the most identical sequences was conducted using the BLAST algorithm [31] from the GISAID database [32].

The obtained sets of sequences were subjected to multiple sequence alignment using the MAFFT v7.520 multiple alignment algorithm (downloadable version) [33] with the following default parameters: gap opening penalty: 1.53, gap extension penalty: 0.0. Then, the files containing multiple alignments were opened in the Unipro UGENE software, version 50.0 [34], where they were manually checked, and sites containing deletions were removed from the final alignment.

Phylogenetic trees were constructed using the IQ-TREE version 1.6 ML (maximum likelihood) method [35]. Branch support was assessed using ultrafast bootstrap v2.4.0 [36] and the SH-aLRT branch test with 1000 iterations. The nucleotide substitution model for each gene segment was determined using ModelFinder through IQ-TREE version 1.6 [37].

Tree visualization and topology analysis were performed using iTOL v.6 [38].

## 3. Results

### 3.1. Sampling and Virus Isolation

As part of the annual influenza A virus monitoring program on the Taimyr Peninsula from 2019 to 2023, 323 samples were collected from wild birds. Samples were obtained from 18 species of four orders: *Charadriiformes* (*n* = 167), *Anseriformes* (*n* = 147), *Gaviiformes* (*n* = 7), and *Passeriformes* (*n* = 2). The most numerous species in the sample were: herring gull (*Larus argentatus*) (*n* = 158), northern pintail (*Anas acuta*) (*n* = 30), european wigeon (*Mareca penelope*) (*n* = 28), and velvet scoter (*Melanitta fusca*) (*n* = 28) (Table 1).

During five years of monitoring, only four APMV isolates and one AIV isolate were detected in the summer and autumn periods. This constituted 1.24% and 0.31% of the total number of samples collected, respectively. All APMV isolates belonged to the *Orthoavulavirus javaense* species. The AIV isolate belonged to the H3N8 subtype.

### 3.2. Phylogenetic Analysis of Newcastle Disease Virus

In the current study, four whole genome sequences of NDV were obtained: NDV/herring gull/Taimyr Peninsula/11k/Russia/2020 (NDV/11k), NDV/herring gull/Taimyr Peninsula/17k/Russia/2020 (NDV/17k), NDV/herring gull/Taimyr Peninsula/18k/Russia/2020 (NDV/18k), and NDV/herring gull/Taimyr Peninsula/19k/Russia/2020 (NDV/19k) (GenBank accession numbers: PV032627-PV032630).

BLAST analysis of the identity of the nucleotide sequences of the whole genome showed that the Taimyr isolates had the highest identity with the isolate we previously obtained from a gull in the south of Eastern Siberia in 2014. Since the GenBank database contains significantly more nucleotide sequences of the F gene of NDV than whole-genome sequences, CDS of the F gene was also used to determine the nucleotide identity. The highest identity (99.46–99.82%) was observed with isolates from China obtained from the black-headed gull, black-tailed gull, and unspecified wild birds in 2018 and 2020 (Table 2).

#### 3.2.1. Gene F

Phylogenetic analysis of the complete CDS gene F (1662 nt) revealed that all Taimyr isolates belonged to genotype I within Class II. When combined with other isolates obtained from various regions of Eurasia, they formed a distinct gull-like clade in sub-genotype I.1.2.1 (see Figure 2). The clade was mainly formed by isolates from different species of gulls: black-tailed gull, black-headed gull, little gull, herring gull, and slaty-backed gull. These isolates were collected sporadically and with long intervals between occurrences from different locations across the continent. Also in the clade was an isolate obtained in the USA from the red knot shorebird, belonging to the same order as gulls, *Charadriformes*. NDV/BHG/Sweden/94 and APMV-1/red knot/US(NJ)/A101-1383/2001, the oldest in the clade, formed a separate branch.

#### 3.2.2. Whole Genome

When comparing the whole genome sequences of NDV, it was seen that a gull-like clade also formed in sub-genotype I.1.2.1 (Figure 3). Currently, the NCBI database has significantly fewer available open-access whole-genome sequences of NDV than sequences of the F gene. Therefore, the gull-like clade on the phylogenetic tree of the whole genome is represented only by isolates from the Taimyr Peninsula, the south of Eastern Siberia, and Sweden.

Isolates used in this study are shown in red. Roman numerals indicate each isolate’s corresponding genotype and sub-genotype according to the classification proposed by Dimitrov et al. [2]. The percentage of trees in which associated taxa clustered together in the bootstrap test (1000 replicates) is shown next to the branches.

#### 3.2.3. Amino Acid Substitutions

Analysis of amino acid substitutions of the F protein showed that the sequences obtained in this study contained the F protein cleavage site 112-GKQGR↓L-117, characteristic of lentogenic virus. Other viruses in sub-genotype I.1.2.1, including vaccine strains, contained the same cleavage site (Table 3). In sub-genotype I.1.2.2, the F protein cleavage site was the same as in isolates of sub-genotype I.1.2.1, but a variation at position 112 was noted in the isolate northern pintail/USA/AK/44500/136/2009.

Sub-genotype I.1.1 contained isolates characteristic of both highly pathogenic and low-pathogenic viruses, reflecting the historical composition of sub-genotype I.1.1 [14]. For sub-genotype I.2, cases of detection of only lentogenic isolates were described, which was confirmed by the corresponding cleavage site.

### 3.3. Phylogenetic Analysis of Avian Influenza Virus

From 2019 to 2023, one H3N8 avian influenza virus, A/northern pintail/Taimyr/Tm27/2023 (EPI_ISL_19202454), was isolated and sequenced. Phylogenetic dendrograms were constructed using genome segment sequences from the isolated influenza virus strain, and closely related sequences from around the world were identified through BLAST analysis.

The H3 segment sequence forms a distinct branch, sharing a common ancestor with European H3 viruses, including strains from Belgium and Germany, and H3N2 strains isolated from a wild duck in the Moscow Pond in 2023 (Figure 4) [39]. This hemagglutinin virus belongs to the classical, avian-like clade of Eurasian low-pathogenic H3Nx viruses [40].

Conversely, based on the neuraminidase segment, the N8 sequence (Figure 5) does not group within this specific European (Belgian–German) clade. Instead, it resides in a more diverse clade of H3N8 viruses. Notably, the clade also contains a unique H5N8 virus that was isolated from a duck in the Netherlands in 2023.

The viruses that share the specific clades of our studied H3 and N8 segments are all contemporary strains circulated between 2021 and 2023. This supports a phylogenetic relationship and suggests that these strains are circulating in northern regions, such as Taimyr, and are likely spread by wild, migrating birds.

Phylogenetic analysis of internal segments showed that all are of Eurasian lineage. Figures are presented in the Appendix A. Interestingly, the polymerase complex segments exhibit distinct phylogenetic relationships: PB1 and PB2 group within European clades, whereas the PA segment clusters with East and Central Asian viruses isolated in Eastern Siberia, Japan, China, and Bangladesh. Additionally, the NS segment of the virus we isolated belongs to allele B, typical for avian influenza viruses, instead of allele A, which is also found in mammalian viruses [41].

Using the FluSurver tool [42], we analyzed genome segment sequences to identify mutations resulting in amino acid substitutions. The identified substitutions are detailed in Appendix A. While some substitutions, such as I262V in neuraminidase, are known to confer increased resistance to antiviral drugs, potentially, all identified substitutions at critical positions in our analysis presented residues different from those previously associated with the described effect. Our findings indicate the presence of substitutions that, based on current knowledge, do not elevate the risk of adaptation to mammals. However, a key limitation is that the functional roles of these substitutions have primarily been characterized in H5N1 influenza viruses and have not yet been definitively shown for low-pathogenic avian influenza viruses (LPAIs).

The phylogenetic analysis revealed no specific pattern for A/northern pintail/Taimyr/Tm27/2023. All segments belong to the Eurasian clades of classic avian-like influenza viruses. The close phylogenetic relationships of the virus with Asian and Siberian viruses on the one hand, and with European viruses on the other, suggest broad connections and ways of virus transmission throughout Eurasia into Arctic regions, as we demonstrated previously for Eurasian regions [43,44,45].

## 4. Discussion

During a five-year study in the summer and autumn periods on the Taimyr Peninsula (Arctic region of Russia), only four NDV isolates and one AIV isolate were detected.

The H3N8 subtype is one of the most commonly detected avian influenza viruses in wild birds, particularly in Northern Asia [44]. While H3N8 viruses are widespread in birds, human infections are rare [46]. However, it is important to monitor these viruses due to their potential to evolve and cause human disease, as demonstrated by isolated cases and a recent fatal case in China. We did not find any substitutions increasing virulence to mammals or segments of highly pathogenic avian influenza lineages, confirming the low pathogenicity of the virus studied.

Polar regions may become new frontiers for the emergence and spread of significant viral pathogens. In Antarctica, mass deaths of penguins of unknown etiology were recorded back in the 1970s [18]. Currently, influenza A virus and Newcastle disease virus have been detected in the region. The first studies reported the detection of antibodies against these viruses in the Adélie penguin (*Pygoscelis adeliae*) and the south polar skua (*Stercorarius maccormicki*) [47,48]. Current studies have confirmed the presence of H5N1 HPAIV in birds and seals, and low-pathogenicity variants of NDV in penguins [21,49]. A study of the spread of H5N1 HPAIV from South America to Antarctica has confirmed the involvement of seals and several bird species: brown skua (*Stercorarius antarcticus*), kelp gull (*Larus domincanus*), and antarctic tern (*Sterna vittata*) [19,20].

A recent comprehensive review of influenza A virus ecology in the Arctic found that Alaska is the best-studied region of the Arctic [22]. There has been very little research on the influenza A virus in the Arctic region of Russia. The only known large-scale surveillance of avian influenza viruses on the Taimyr Peninsula, which involved testing over 1000 geese of various species during their spring migration, did not detect any avian influenza viruses [50]. The Arctic region of Russia includes subarctic, low Arctic, and high Arctic zones with a variety of wetlands and climate conditions that attract different species of waterbirds. The introduction of viruses into relatively isolated polar regions with low species diversity could have detrimental effects. More research in this region will contribute to our understanding of the genetic flow and distribution of avian influenza viruses.

We focused on analyzing unique NDV isolates obtained from herring gulls in autumn 2020. As a result of phylogenetic analysis, these isolates and other isolates from gulls formed a strongly supported clade in sub-genotype I.1.2.1, Class II.

Genotype I is one of the four “historical” genotypes, currently represented by four sub-genotypes: I.1.1, I.2, I.1.2.1, and I.1.2.2 [2]. Viruses of sub-genotypes I.2 and I.1.1 have been detected in wild and domestic birds worldwide. Isolates of sub-genotype I.2 are low-pathogenic, while sub-genotype I.1.1 also includes highly pathogenic isolates [14]. These sub-genotypes also include the vaccine strains Queensland/V-4/chicken/Australia/1966 and Ulster/chicken/Ireland/1967. The isolates that formed the I.1.2.2 sub-genotype were isolated from waterfowl, shorebirds, and redpoll from 2002 to 2009 and from poultry in Japan in 1962.

Sub-genotype I.1.2.1 is represented by low-pathogenic isolates from wild and domestic birds and vaccine strains NDV/AVIVAK-NDV-BOR/74/2022 (used in Russia) and NDV/PHY-LMV42/Hungary. Viruses from domestic birds were detected in Ukraine in 1992 and Vietnam in 2015. Viruses from wild birds (mostly from gulls) were detected in Sweden in 1994, the USA in 2001, Japan in 2009, and Russia in 2014. Recently, sequences from China, obtained from two gull species and unidentified wild birds in 2018 and 2020, were added to the NCBI database. Together with the sequences obtained during this study, they formed a separate gull-like clade that has not been previously identified.

The gull-like clade is represented by sequences obtained in different regions and with long intervals between occurrences. The oldest isolate in the clade was obtained from a black-headed gull in Sweden in 1994. The sample was collected in the autumn at the Ottenby Bird Observatory, on the Baltic coast [51]. The sample included 430 samples from 57 bird species, but no other data on APMV detection in this sample were described. Retrospective and ongoing analyses of samples from 1986 to 2006 in the United States yielded 268 NDV isolates from wild waterbirds and live bird market birds [6]. Among them, only one isolate, obtained from a red knot in 2001, was most closely related to the isolate from Sweden. During monitoring of AIV and APMV in migratory birds in Alaska, Russia, and Japan from 2007 to 2010, 73 NDV isolates were obtained. Only one isolate, obtained from a slaty-backed gull in Japan in 2009, was phylogenetically close to isolates from Sweden and the USA, which led to the identification of a new sub-genotype I.1.2.1 (formerly 1c) [52]. In 2014, our laboratory staff obtained a full-genome sequence of NDV from a little gull due to cultivation. Biological material from the bird was collected in April at Ubsu-Nur Lake on the border of Russia and Mongolia. From 2014 to 2021, we did not detect any other APMV isolates in this region. In 2018 and 2020, several NDV sequences were obtained from a black-headed gull, a black-tailed gull, and unspecified wild bird species in China. Seven of the ten isolates were isolated in March 2020, two in November 2020, and one in September 2018. All were phylogenetically related to the gull isolates described above. These sequences from unspecified species may belong to gulls or other species of the order *Charadriiformes*. However, there is no precise information. Identifying the host bird species or genus during sample collection is important for monitoring viruses with relevance for the poultry industry. Detection of the affiliation of a specific viral lineage to a particular host species helps to monitor the direction of virus evolution and adjust preventive measures.

In the current study, we found NDV in four herring gulls. This is a common bird species on the Taimyr Peninsula. These birds arrive in the spring and build nests on islands and along the shores of large lakes and rivers, forming colonies. The first spring migrations to the Taimyr Peninsula begin in late April. The bulk of nesting sites for this species are concentrated in the west of the peninsula (the Yenisei River floodplain) and the southeast (the foothills of the Putorana Plateau). Birds migrate to the eastern part of the peninsula from the south and southeast along the Kotuy, Popigai, and Anabar Rivers. Some flocks of herring gulls fly to the shores of Taimyr and Severnaya Zemlya from the Laptev Sea, from the New Siberian Islands. Autumn migration to wintering grounds begins in the last ten days of August and continues until the end of September. However, in the area of city dumps, which are located within the boundaries of the Norilsk industrial region, large aggregations of herring gulls linger for a longer period: until the end of the first ten days of October and even later. They winter in Japan, Korea, and China [53]. Within Russia, they winter on the eastern coast of the Baltic Sea, the northern coast of the Black and Azov Seas, the Caspian Sea, in the south of Primorsky Krai, and in the Kuril Islands [54]. Thus, the ecological characteristics of the species make it possible for viruses to spread not only within the Central Asian and East Asian Flyways, but also in areas of large bird concentrations at the intersection of several flyways (for example, on the western coast of the Caspian Sea).

The NDV isolates obtained in this study contained the F protein cleavage site characteristic of low-virulence viruses [55]. Samples were collected from asymptomatic gulls. Avirulent isolates cause subclinical infections or mild respiratory disease and are not important for agri-livestock farming, but their role in the evolution of NDV remains to be elucidated. In a recent study, avirulent NDV isolates were obtained during an outbreak of highly pathogenic avian influenza in the UK [56]. Avirulent NDV was isolated from both a chicken in one of the poultry houses and the carcasses of a magpie (*Pica pica*) found near the poultry houses. Co-infection of one organism with several virus species is associated with the emergence of complex interspecific relationships under selection pressure [57]. Thus, avirulent isolates may contribute more to the formation of the overall genetic landscape of NDV.

Based on the results of phylogenetic analysis, the gull isolates obtained in this study in 2020 are most closely related to the isolates obtained in China in 2018 and 2020 and in the south of Eastern Siberia (Ubsu-Nur Lake, Russia) in 2014. The nucleotide identity of the F gene was 97.35–99.82%. This may indicate the exchange of viruses within the Central Asian and East Asian Flyways. Given that gull NDV sequences, which were obtained sporadically and with long intervals between occurrences from different locations across the continent, formed a distinct clade in sub-genotype I.1.2.1, we suggest that within Class II there may be a larger number of genetic lineages leading to adaptation to a larger number of hosts.

The diversity of susceptible hosts expands the range of conditions for the emergence of different quasispecies during the virus life cycle. A study conducted in 2010 showed that virulent viral populations can arise during the replication of avirulent field isolates [58]. The presence of such virulent populations did not lead to a change in the phenotypic properties of the avirulent isolate. However, the selective pressure caused by passaging led to an increase in the percentage of virulent quasispecies. The authors believe that the selection pressure can lead to the dominance of virulent quasi-species and, as a consequently, to a change in phenotypic properties. The diversity of susceptible hosts with a rapid shift in generations and individual biological characteristics can also change the parameters of immune pressure, thereby changing the direction of selection in the evolution of the virus. The existence of a lineage of NDV specific to species from the order *Charadriiformes* can be accompanied by the accumulation of mutations arising under the influence of population immunity. If the biological features that influence population immunity differ significantly between the orders *Anseriformes* and *Charadriiformes*, then virus specialization to *Charadriiformes* species will increase the diversity of the overall gene pool that is subject to evolutionary events. Further surveillance of different gull and shorebird species may help better understand their role in the evolution and global spread of NDV.

NDV and H3N8 isolates were obtained from herring gulls and northern pintails. The northern pintail is recognized as a reservoir for avian influenza virus and can carry the virus asymptomatically, playing an important role in long-distance transmission along migratory routes [59,60]. Newcastle disease virus has also been detected in clinically healthy northern pintails [61]. Northern pintail migrations include transcontinental routes, so it is important to monitor for viral pathogens in these bird populations. Herring gulls are opportunistic scavengers and have been implicated in the spread of avian pathogens. Their ecological behavior may be key to the spread of the virus. For example, a similar species, the kelp gull (*Larus dominicanus*), has been documented scavenging on elephant seal (*Mirounga leonina*) carcasses, highlighting the potential for interspecies transmission and ecological persistence of viral pathogens [62].

Nevertheless, our five-year study has identified certain limitations that should be acknowledged. The main limitation is that the sample size was small (*n* = 323). In addition, the study included 48.92% of samples from the herring gull, which may introduce bias. However, we wanted to estimate the general viral diversity in a poorly studied area. *Charadriiformes* (51.7%) and *Anseriformes* (45.51%) are informative groups for an initial assessment of viral diversity.

## 5. Conclusions

In this study, we report the first detection of Newcastle disease virus (NDV) in wild birds in Arctic Russia. All NDV sequences were obtained from the herring gull and, together with other gull sequences, formed a separate gull-like clade in the sub-genotype I.1.2.1, Class II. The gull isolates obtained in this study in 2020 are closely related to those isolates obtained in China in 2018 and 2020 and in the south of Eastern Siberia (Russia) in 2014. The nucleotide identity of the F gene was 97.35–99.82%. This may indicate the exchange of viruses within the Central Asian and East Asian Flyways.

Our study also identified an avian influenza virus subtype H3N8 isolated from a northern pintail. This isolate was phylogenetically related to viruses circulating between 2021 and 2023 in Eurasia, Siberia, and Asia.

Further research in this region will further our understanding of the distribution and evolution of avian paramyxoviruses and avian influenza viruses.

## Figures and Tables

**Figure 1 viruses-17-00955-f001:**
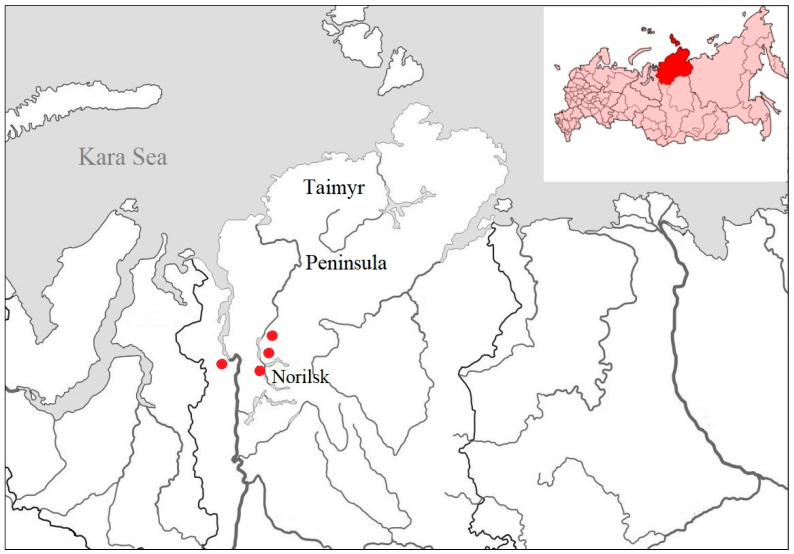
Sampling sites in the Taimyr Peninsula (Krasnoyarsk region, Russia).

**Figure 2 viruses-17-00955-f002:**
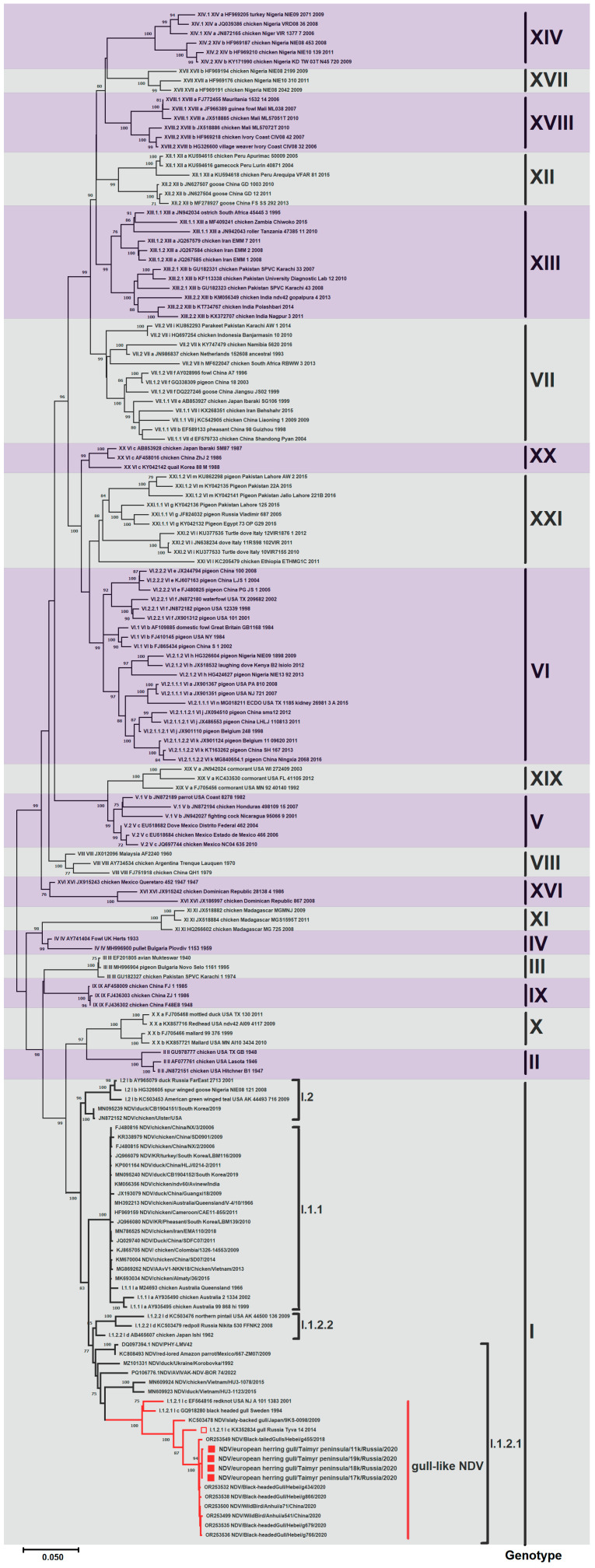
Maximum likelihood phylogenetic tree of the F gene (1662 nt) of NDV Class II (■—sequence from the study; □—sequence we obtained in a previous study). The branches of the tree that form the gull-like clade are colored in red. Isolates used in this study are shown in red. Roman numerals indicate each isolate’s corresponding genotype and sub-genotype according to the classification proposed by Dimitrov et al. [2]. The percentage of trees in which the associated taxa clustered together in the bootstrap test (1000 replicates) is shown next to the branches.

**Figure 3 viruses-17-00955-f003:**
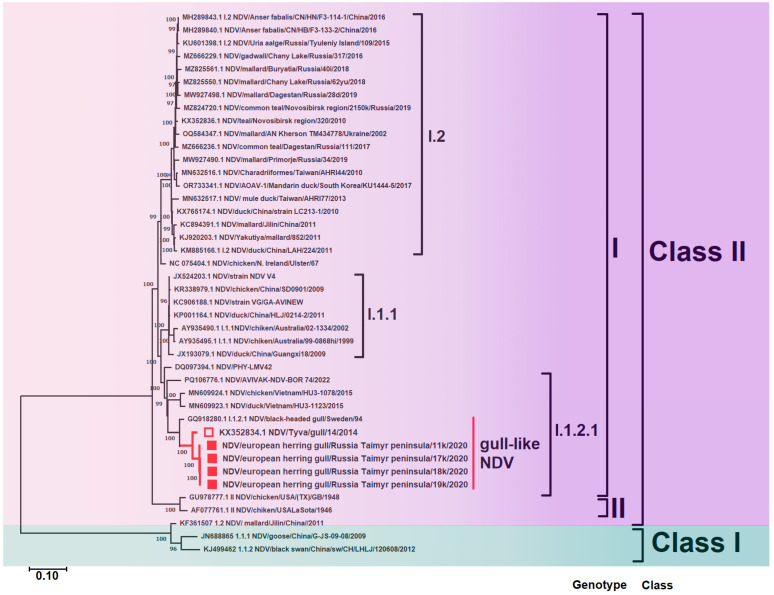
Maximum likelihood phylogenetic tree of the whole genome of NDV Class II (■—sequence from the study; □—sequence we obtained in a previous study).

**Figure 4 viruses-17-00955-f004:**
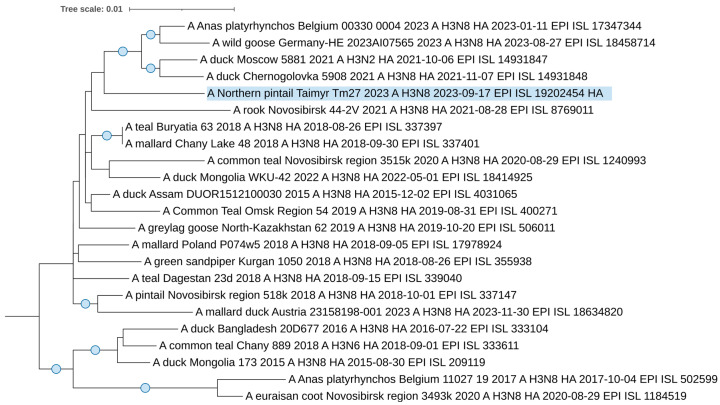
Maximum likelihood phylogenetic tree of the HA (H3) genome segment of avian influenza viruses isolated in the Taimyr Peninsula. The blue circle symbol denotes branches with values SH-aLRT > 80% and UFboot > 95%.

**Figure 5 viruses-17-00955-f005:**
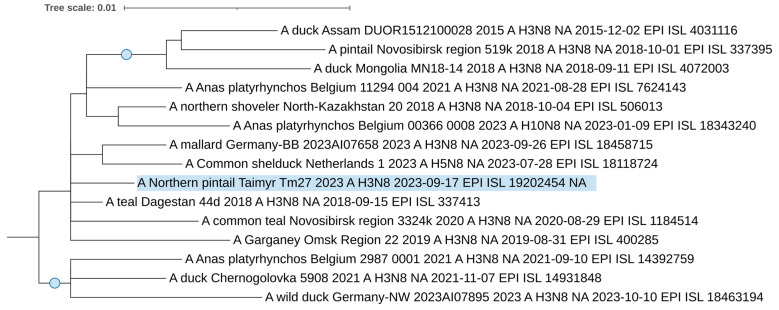
Maximum likelihood phylogenetic tree of the NA (N8) genome segment of avian influenza viruses isolated in the Taimyr Peninsula. The blue circle symbol denotes branches with values SH-aLRT > 80% and UFboot > 95%.

**Table 1 viruses-17-00955-t001:** Composition and results of virus detection in wild birds of the Taimyr Peninsula, 2019–2023.

Order	Family	Species	Total Number	Number of NDV	Number of AIV
**Anseriformes**	Anatidae	Greater white-fronted goose (*Anser albifrons*)	9	0	0
12 species (*n* = 147)	Anatidae	Common merganser (*Mergus merganser*)	4	0	0
	Anatidae	Long-tailed duck (*Clangula hyemalis*)	15	0	0
	Anatidae	Common goldeneye (*Bucephala clangula*)	7	0	0
	Anatidae	Taiga bean-goose (*Anser fabalis*)	1	0	0
	Anatidae	Eurasian wigeon (*Mareca penelope*)	28	0	0
	Anatidae	Velvet scoter (*Melanitta fusca*)	28	0	0
	Anatidae	Gadwall (*Mareca strepera*)	1	0	0
	Anatidae	Greater scaup (*Aythya marila*)	18	0	0
	Anatidae	Tufted duck (*Aythya fuligula*)	2	0	0
	Anatidae	Green-winged teal (*Anas crecca*)	4	0	0
	Anatidae	Northern pintail (*Anas acuta*)	30	0	**1**
**Charadriiformes**	Stercorariidae	Long-tailed jaeger (*Stercorarius longicaudus*)	2	0	0
4 species (*n* = 167)	Charadriidae	European golden-plover (*Pluvialis apricaria*)	4	0	0
	Scolopacidae	Ruff (*Calidris pugnax*)	3	0	0
	Laridae	Herring gull (*Larus argentatus*)	158	**4**	0
**Gaviiformes** 1 species (*n* = 7)	Gaviidae	Arctic loon (*Gavia arctica*)	7	0	0
**Passeriformes** 1 species (*n* = 2)	Corvidae	Hooded crow (*Corvus cornix*)	2	0	0
**Total**	**7**	**18**	**323**	**4**	**1**

**Table 2 viruses-17-00955-t002:** Nucleotide identity of Taimyr isolates.

Strain	Host	Related Strain	Country	Host	Whole-Genome Identity, %	Gene F Identity, %
NDV/11k	*Larus* *argentatus*	APMV-1/WildBird/Anhui/a71/2020	China	Wild bird	- ^1^	99.76
		APMV-1/Black-headedGull/Hebei/g679/2020	China	*Chroicocephalus ridibundus*	-	99.70
		NDV/Tyva/gull/14/2014	Russia	*Hydrocoloeus minutus*	97.30	97.29
		APMV-1/slaty-backed gull/Japan/9KS-0098/2009	Japan	*Larus schistisagus*	-	96.63
		APMV-1/red knot/US(NJ)/A101-1383/2001	USA	*Calidris canutus*	-	94.04
		NDV/BHG/Sweden/94	Sweden	*Chroicocephalus ridibundus*	94.44	93.98
		PHY-LMV42	Hungary	Vaccine strain	91.41	91.16
		AVIVAK-NDV-BOR 74	Russia	Vaccine strain	90.72	91.82
		NDV/chicken/Vietnam/HU3-1078/2015	Vietnam	Chicken	90.31	90.55
NDV/17k	*Larus* *argentatus*	APMV-1/WildBird/Anhui/a71/2020	China	Wild bird	-	99.82
		APMV-1/Black-headedGull/Hebei/g679/2020	China	*Chroicocephalus ridibundus*	-	99.76
		NDV/Tyva/gull/14/2014	Russia	*Hydrocoloeus minutus*	97.24	97.35
		APMV-1/slaty-backed gull/Japan/9KS-0098/2009	Japan	*Larus schistisagus*	-	96.69
		APMV-1/red knot/US(NJ)/A101-1383/2001	USA	*Calidris canutus*	-	94.10
		NDV/BHG/Sweden/94	Sweden	*Chroicocephalus ridibundus*	94.40	94.04
		PHY-LMV42	Hungary	Vaccine strain	91.37	91.16
		AVIVAK-NDV-BOR 74	Russia	Vaccine strain	90.68	91.88
		NDV/chicken/Vietnam/HU3-1078/2015	Vietnam	Chicken	90.30	90.61
NDV/18k	*Larus* *argentatus*	APMV-1/WildBird/Anhui/a71/2020	China	Wild bird	-	99.82
		APMV-1/Black-headedGull/Hebei/g679/2020	China	*Chroicocephalus ridibundus*	-	99.76
		NDV/Tyva/gull/14/2014	Russia	*Hydrocoloeus minutus*	97.23	97.35
		APMV-1/slaty-backed gull/Japan/9KS-0098/2009	Japan	*Larus schistisagus*	-	96.69
		APMV-1/red knot/US(NJ)/A101-1383/2001	USA	*Calidris canutus*	-	94.10
		NDV/BHG/Sweden/94	Sweden	*Chroicocephalus ridibundus*	94.39	94.04
		PHY-LMV42	Hungary	Vaccine strain	91.36	91.22
		AVIVAK-NDV-BOR 74	Russia	Vaccine strain	90.68	91.88
		NDV/chicken/Vietnam/HU3-1078/2015	Vietnam	Chicken	90.29	90.55
NDV/19k	*Larus* *argentatus*	APMV-1/WildBird/Anhui/a71/2020	China	Wild bird	-	99.82
		APMV-1/Black-headedGull/Hebei/g679/2020	China	*Chroicocephalus ridibundus*	-	99.76
		NDV/Tyva/gull/14/2014	Russia	*Hydrocoloeus minutus*	97.22	97.35
		APMV-1/slaty-backed gull/Japan/9KS-0098/2009	Japan	*Larus schistisagus*	-	96.69
		APMV-1/red knot/US(NJ)/A101-1383/2001	USA	*Calidris canutus*	-	94.10
		NDV/BHG/Sweden/94	Sweden	*Chroicocephalus ridibundus*	94.39	94.04
		PHY-LMV42	Hungary	Vaccine strain	91.35	91.22
		AVIVAK-NDV-BOR 74	Russia	Vaccine strain	90.67	91.88
		NDV/chicken/Vietnam/HU3-1078/2015	Vietnam	Chicken	90.29	90.61

^1^ Complete genome sequences are not available.

**Table 3 viruses-17-00955-t003:** Amino acid substitutions of the F protein of NDV.

Strain	Cleavage Site Amino acid Sequence	Year of Isolation	Sub-Genotype	GenBank Accession No.
BHG/Sweden/94	^112^GKQGR↓L^117^	1994	I.1.2.1	GQ918280
Slaty-backed gull/Japan/9KS-0098/2009		2009	I.1.2.1	KC503478
Tyva/gull/14/2014		2014	I.1.2.1	KX352834
Black-headedGull/Hebei/g679/2020		2020	I.1.2.1	OR253535
WildBird/Anhui/a71/2020		2020	I.1.2.1	OR253500
Red knot/USA/NJ/A101/1383/2001		2001	I.1.2.1	EF564816
NDV/11k		2020	I.1.2.1	PV032627
NDV/17k		2020	I.1.2.1	PV032628
NDV/18k		2020	I.1.2.1	PV032629
NDV/19k		2020	I.1.2.1	PV032630
NDV/chicken/Vietnam/HU3-1078/2015		2015	I.1.2.1	MN609924
AVIVAK-NDV-BOR 74		2022	I.1.2.1	PQ106776
PHY-LMV42			I.1.2.1	DQ097394
Northern pintail/USA/AK/44500/136/2009	^112^**E**KQGR↓L^117^	2009	I.1.2.2	KC503476
Chicken/Japan/Ishi/1962	^112^GKQGR↓L^117^	1962	I.1.2.2	AB465607
Chicken/Australia/Queensland/V-4/10/1966	^112^GKQGR↓L^117^	1966	I.1.1	MH392213
Chicken/Australia/2/1334/2002	^112^**RR**QR**R**↓**F**^117^	2002	I.1.1	AY935490
Chicken/Australia99/86 hi/1999	^112^**RR**QGR↓L^117^	1999	I.1.1	AY935495
Chicken/Ulster/USA	^112^GKQGR↓L^117^	1967	I.2	JN872152
Duck/Russia/FarEast/2713/2001		2001	I.2	AY965079

## Data Availability

All sequences from the study are available in the GenBank database (accession numbers: PV032627-PV032630) and in the GISAID EpiFlu database (accession numbers: EPI_ISL_19202454).

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
