# Peer review of "Detection of a Novel Gull-like Clade of Newcastle Disease Virus and H3N8 Avian Influenza Virus in the Arctic Region of Russia (Taimyr Peninsula)"

_viruses, 2025, doi:10.3390/v17070955_

Round 1
Reviewer 1 Report
Comments and Suggestions for Authors
Manuscript ID: viruses-3688977_Type of manuscript: Article_Title: Detection of Avian influenza virus and a Novel Gull-like clade 2 of Newcastle Disease Virus in the Arctic region of Russia (Taimyr Peninsula)
COMMENTS TO THE AUTHORS
Brief summary
From 2019 to 2023, during influenza A virus monitoring of wild birds sampled in the Taimyr Peninsula (Krasnoyarsk region, Russia), whole-genome sequences of four Newcastle disease virus (NDV) isolates and one H3N8 avian influenza virus (AIV) isolate were obtained from 323 samples collected from wild birds.
Broad comments
This study reports the first detection of NDV in wild birds sampled in the Arctic Russia. All the four NDV sequences obtained from the Herring gull clustered together with other gull sequences, forming a separate gull-like clade in the sub-genotype I.1.2.1, Class II. The gull isolates under study, obtained in 2020, were closely related to isolates obtained in China in 2018 and 2020 and in the south of Eastern Siberia (Russia) in 2014. High values of nucleotide identity of the F gene may indicate possible exchange of NDVs within the Central Asian and East Asian Flyways.
The H3N8 AIV, isolated from a Northern pintail, was phylogenetically related to viruses circulating between 2021 and 2023 in Eurasia, Siberia, and Asia.
As stated by the Authors themselves, further research in this region could allow us to better understand the distribution and evolution of avian paramyxoviruses and avian influenza viruses, including their host adaptation.
- In general, I recommend a few small changes which could help to improve the manuscript clarity.
Please, see “Specific comments” for details.
Specific comments:
Title:
In my opinion, given the data produced, the manuscript title could be improved. I suggest adding H3N8 and possibly to rephrase it, e.g. as follows: “Detection of a Novel Gull-like clade 2 of Newcastle Disease Virus and H3N8 Avian influenza virus in the Arctic region of Russia (Taimyr Peninsula)”
- Pag. 1_24: “..... detection of in Arctic Russia .....”, “of” can be deleted.
Keywords:
- I suggest to add Northern pintail in the keyword list.
Introduction:
- Pag. 2_60: “ ….. That are important for agriculture and …..”. Did you mean to say that these viruses may impact the agri-livestock farming system and biodiversity conservation? I suggest integrating the sentence.
- Pag. 3_83: 2 ml of each embryo” ….. Did you mean to say 2 ml of allantoic fluid from each embryo …..? If necessary, please rephrase this sentence.
Results:
- Pag. 4-5, Table 1: Please use italic font for the genus and species of birds. The number of bird families is 7, not 6. Please correct it in the table.
In addition, I found some differences between bird names you used and those reported on this site https://birdsoftheworld.org/bow/home . These differences are shown by the following underlined terms: Eurasian wigeon (Mareca penelope); Gadwall (Mareca strepera); Green-winged teal (Anas crecca); Ruff (Calidris pugnax); Arctic loon (Gavia arctica). In general, I suggest using updated classification criteria.
- Pag. 6_168: Please amend the conjunction in Russian.
- Pag. 5-6, Table 2: In the “Whole genome identity %” column, please replace the comma with a point to indicate decimals.
- Pag. 7_188: “ ….. was an isolated isolate in the USA …..” could be replaced with “ …. was an isolate obtained in the USA …..”.
Discussion:
- 12_334-335: “….. agriculturally significant viruses …..”. Please see the Pag. 2_60 comment.
- 13_359-360: “….. not significant for agriculture …..”. Please see the Pag. 2_60 comment.
- 13_361: “ ….. NDV isolates were isolated …..” could be replaced with “ ….. NDV isolates were obtained …..”
Supplementary Materials:
In Figures S4 and S6, it would be better to increase the space between the captions and phylogenetic trees.
Author Response
Dear Reviewers, Editor,
Thank you all for your useful comments and constructive suggestions which helped us to improve the manuscript. Additional information was provided for introduction, results and discussion according to the comments. We edited/modified study and text significantly according to comments and suggestions.
Please see below detailed response on each point of the review and corrections of the manuscript.
Comment 1: Title: In my opinion, given the data produced, the manuscript title could be improved. I suggest adding H3N8 and possibly to rephrase it, e.g. as follows: “Detection of a Novel Gull-like clade of Newcastle Disease Virus and H3N8 Avian influenza virus in the Arctic region of Russia (Taimyr Peninsula)”
Response 1: We agree with the reviewer’s comment. Thank you for your helpful comments. Corrected according to the comment.
Comment 2: Pag. 1_24: “..... detection of in Arctic Russia .....”, “of” can be deleted.
Response 2: corrected according to the comment.
Comment 3: Keywords: I suggest to add Northern pintail in the keyword list.
Response 3: corrected according to the comment.
Comment 4: Pag. 2_60: “ ….. That are important for agriculture and …..”. Did you mean to say that these viruses may impact the agri-livestock farming system and biodiversity conservation? I suggest integrating the sentence.
Response 4: corrected according to the comment ("Insufficient information on infectious agents in the Arctic limits understanding of the genetic diversity and evolution of viruses having potential impact on the agri-livestock farming and especially the poultry industry and on the species biodiversity conservation in general").
Comment 5: Pag. 3_83: 2 ml of each embryo” ….. Did you mean to say 2 ml of allantoic fluid from each embryo …..? If necessary, please rephrase this sentence.
Response 5: corrected according to the comment.
Comment 6: Pag. 4-5, Table 1: Please use italic font for the genus and species of birds. The number of bird families is 7, not 6. Please correct it in the table. In addition, I found some differences between bird names you used and those reported on this site https://birdsoftheworld.org/bow/home . These differences are shown by the following underlined terms: Eurasian wigeon (Mareca penelope); Gadwall (Mareca strepera); Green-winged teal (Anas crecca); Ruff (Calidris pugnax); Arctic loon (Gavia arctica). In general, I suggest using updated classification criteria.
Response 6: corrected to meet the updated classification criteria.
Comment 7: Pag. 6_168: Please amend the conjunction in Russian.
Response 7: corrected according to the comment.
Comment 8: Pag. 5-6, Table 2: In the “Whole genome identity %” column, please replace the comma with a point to indicate decimals.
Response 8: corrected according to the comment.
Comment 9: Pag. 7_188: “ ….. was an isolated isolate in the USA …..” could be replaced with “ …. was an isolate obtained in the USA …..”.
Response 9: corrected according to the comment.
Comment 10: 12_334-335: “….. agriculturally significant viruses …..”. Please see the Pag. 2_60 comment.
Response 10: corrected according to the comment ("Identifying of the host bird species or genus during sample collection is important in monitoring viruses with relevance for the poultry industry").
Comment 11: 13_359-360: “….. not significant for agriculture …..”. Please see the Pag. 2_60 comment.
Response 11: corrected according to the comment ("Avirulent isolates cause subclinical infections or mild respiratory disease and are not important for the agri-livestock farming, but their role in the evolution of NDV remains to be elucidated").
Comment 12: 13_361: “ ….. NDV isolates were isolated …..” could be replaced with “ ….. NDV isolates were obtained …..”
Response 12: corrected according to the comment.
Comment 13: In Figures S4 and S6, it would be better to increase the space between the captions and phylogenetic trees.
Response 13: corrected according to the comment.
Reviewer 2 Report
Comments and Suggestions for Authors
The manuscript submitted by Derko et al., entitled “Detection of Avian Influenza Virus and a Novel Gull-like Clade 2 of Newcastle Disease Virus in the Arctic Region of Russia (Taimyr Peninsula)”, presents the detection and genomic characterization of avian influenza virus (AIV) and Newcastle disease virus (NDV) in wild waterbirds from the Taimyr Peninsula in Arctic Russia. This study fills an important knowledge gap in viral surveillance in Arctic breeding grounds and provides valuable data on the phylogenetic position of NDV and H3N8 strains detected in gulls. The authors analyzed a substantial number of samples and conducted a meaningful surveillance effort in a remote and logistically challenging region. Although the number of positive samples was limited, their detection highlights the importance of this work, particularly given that both viruses identified are associated with economic losses in commercial poultry production and negative impacts on wildlife health. The manuscript is timely and relevant, especially in the context of viral evolution, host adaptation, and the role of migratory birds in the global dissemination of pathogens. It is suitable for publication in Viruses pending minor revision.
Minor revisions:
1) The authors should provide a brief overview of the influenza virus in the Introduction section, including its taxonomic classification. Specifically, they should refer to its current scientific name, Alphainfluenzavirus influenzae, as established in recent taxonomic updates. A suitable reference for this classification is: Relich RF, Loeffelholz MJ. Taxonomic Changes for Human Viruses, 2020 to 2022. J Clin Microbiol. 2023 Jan 26;61(1):e0033722. doi: 10.1128/jcm.00337-22.
2) The Introduction should include a paragraph acknowledging that wild birds are not the sole drivers of avian virus dissemination. It is essential to note that the commercial poultry trade also plays a significant role in the global spread of avian influenza viruses. As emphasized by Kilpatrick et al. (2006), “These results highlight the potential synergism between trade and wild animal movement in the emergence and pandemic spread of pathogens.” Including this perspective would strengthen the epidemiological framing of the manuscript and provide a more balanced understanding of the transmission dynamics. The following reference should be cited: Kilpatrick AM et al. Predicting the global spread of H5N1 avian influenza. Proc Natl Acad Sci U S A. 2006 Dec 19;103(51):19368–73. doi: 10.1073/pnas.0609227103.
3) The manuscript would benefit from a broader ecological and geographical context by drawing a comparison between the current detection of avian influenza virus in the Arctic and the recent arrival of highly pathogenic avian influenza (HPAI) H5N1 in the Antarctic region. Including this perspective in both the Introduction and Discussion sections would underscore the significance of polar regions as emerging fronts in the global spread of AIVs, and illustrate how both ends of the globe are increasingly affected. The authors are encouraged to cite and briefly discuss the following recent studies:
- Banyard AC et al. Detection and spread of high pathogenicity avian influenza virus H5N1 in the Antarctic Region. Nat Commun. 2024;15(1):7433. doi: 10.1038/s41467-024-51490-8.
- Kuiken T et al. Emergence, spread, and impact of high-pathogenicity avian influenza H5 in wild birds and mammals of South America and Antarctica. Conserv Biol. 2025; e70052. doi: 10.1111/cobi.70052.
- Ogrzewalska M et al. Genomic analysis of high pathogenicity avian influenza viruses from Antarctica reveals multiple introductions from South America. Research Square Preprint, June 2025. doi: 10.21203/rs.3.rs-6727501/v1
4) The authors should explicitly discuss the ecological characteristics and known roles of the bird species in which AIV and NDV were detected. For example, Northern pintails (Anas acuta) are recognized reservoirs of AIV and may carry the virus asymptomatically, playing a key role in long-distance transmission along migratory flyways. Additionally, Herring gulls (Larus argentatus), identified here as positive for NDV, are opportunistic scavengers and have been implicated in the dissemination of avian pathogens. Their ecological behavior should be discussed in light of recent findings by Uhart et al. (2024), who documented kelp gulls (Larus dominicanus) scavenging on elephant seal carcasses, illustrating the potential for cross-species transmission and environmental persistence of viral pathogens. Including this ecological and epidemiological context would enhance the interpretation of the results and strengthen the manuscript’s relevance to avian disease ecology and transmission dynamics. The following reference should be cited: Uhart MM et al. Epidemiological data of an influenza A/H5N1 outbreak in elephant seals in Argentina indicate mammal-to-mammal transmission. Nat Commun. 2024 Nov 11;15(1):9516. doi: 10.1038/s41467-024-53766-5.
5) The authors should explicitly acknowledge the limitations of the study in the Discussion section. This may include potential sampling bias toward certain species or locations, limited temporal coverage despite the multi-year effort, and the absence of clinical data to assess the pathogenicity or health status of the infected birds. Additionally, the relatively low number of positive detections may limit broader epidemiological conclusions.
Author Response
Dear Reviewers, Editor,
Thank you all for your useful comments and constructive suggestions which helped us to improve the manuscript. Additional information was provided for introduction, results and discussion according to the comments. We edited/modified study and text significantly according to comments and suggestions.
Please see below detailed response on each point of the review and corrections of the manuscript.
Comment 1: The authors should provide a brief overview of the influenza virus in the Introduction section, including its taxonomic classification. Specifically, they should refer to its current scientific name, Alphainfluenzavirus influenzae, as established in recent taxonomic updates. A suitable reference for this classification is: Relich RF, Loeffelholz MJ. Taxonomic Changes for Human Viruses, 2020 to 2022. J Clin Microbiol. 2023 Jan 26;61(1):e0033722. doi: 10.1128/jcm.00337-22.
Response 1: corrected according to the comment (Pag. 1, lines 53-54).
Comment 2: The Introduction should include a paragraph acknowledging that wild birds are not the sole drivers of avian virus dissemination. It is essential to note that the commercial poultry trade also plays a significant role in the global spread of avian influenza viruses. As emphasized by Kilpatrick et al. (2006), “These results highlight the potential synergism between trade and wild animal movement in the emergence and pandemic spread of pathogens.” Including this perspective would strengthen the epidemiological framing of the manuscript and provide a more balanced understanding of the transmission dynamics. The following reference should be cited: Kilpatrick AM et al. Predicting the global spread of H5N1 avian influenza. Proc Natl Acad Sci U S A. 2006 Dec 19;103(51):19368–73. doi: 10.1073/pnas.0609227103.
Response 2: we agree with the reviewer’s comment. Thank you for your helpful comments. Corrected according to the comment (Pag. 1, lines 53-63).
Comment 3: The manuscript would benefit from a broader ecological and geographical context by drawing a comparison between the current detection of avian influenza virus in the Arctic and the recent arrival of highly pathogenic avian influenza (HPAI) H5N1 in the Antarctic region. Including this perspective in both the Introduction and Discussion sections would underscore the significance of polar regions as emerging fronts in the global spread of AIVs, and illustrate how both ends of the globe are increasingly affected. The authors are encouraged to cite and briefly discuss the following recent studies: Banyard AC et al. Detection and spread of high pathogenicity avian influenza virus H5N1 in the Antarctic Region. Nat Commun. 2024;15(1):7433. doi: 10.1038/s41467-024-51490-8.
Kuiken T et al. Emergence, spread, and impact of high-pathogenicity avian influenza H5 in wild birds and mammals of South America and Antarctica. Conserv Biol. 2025; e70052. doi: 10.1111/cobi.70052.
Ogrzewalska M et al. Genomic analysis of high pathogenicity avian influenza viruses from Antarctica reveals multiple introductions from South America. Research Square Preprint, June 2025. doi: 10.21203/rs.3.rs-6727501/v1.
Response 3: we reanalyzed the available literature and used the relevant references suggested by the reviewer. Thank you for your helpful comment. Corrected according to the comment (Pag. 1, lines 53-63; 68-76 and Pag. 12, lines 308-317).
Comment 4: The authors should explicitly discuss the ecological characteristics and known roles of the bird species in which AIV and NDV were detected. For example, Northern pintails (Anas acuta) are recognized reservoirs of AIV and may carry the virus asymptomatically, playing a key role in long-distance transmission along migratory flyways. Additionally, Herring gulls (Larus argentatus), identified here as positive for NDV, are opportunistic scavengers and have been implicated in the dissemination of avian pathogens. Their ecological behavior should be discussed in light of recent findings by Uhart et al. (2024), who documented kelp gulls (Larus dominicanus) scavenging on elephant seal carcasses, illustrating the potential for cross-species transmission and environmental persistence of viral pathogens. Including this ecological and epidemiological context would enhance the interpretation of the results and strengthen the manuscript’s relevance to avian disease ecology and transmission dynamics. The following reference should be cited: Uhart MM et al. Epidemiological data of an influenza A/H5N1 outbreak in elephant seals in Argentina indicate mammal-to-mammal transmission. Nat Commun. 2024 Nov 11;15(1):9516. doi: 10.1038/s41467-024-53766-5.
Response 4: Thank you so much for such a valuable comment! The inclusion of this ecological and epidemiological addition to the text significantly improved the interpretation of the results and increased the importance of the manuscript for studying the ecology of avian diseases and the dynamics of their spread. Corrected according to the comment (Pag. 14, lines 430-440).
Comment 5: The authors should explicitly acknowledge the limitations of the study in the Discussion section. This may include potential sampling bias toward certain species or locations, limited temporal coverage despite the multi-year effort, and the absence of clinical data to assess the pathogenicity or health status of the infected birds. Additionally, the relatively low number of positive detections may limit broader epidemiological conclusions.
Response 5: Thank you for your helpful comment. Corrected according to the comment (Pag. 15, lines 441-446).